# Biological Response to Bioinspired Microporous 3D-Printed Scaffolds for Bone Tissue Engineering

**DOI:** 10.3390/ijms23105383

**Published:** 2022-05-11

**Authors:** Mario Ledda, Miriam Merco, Antonio Sciortino, Elisa Scatena, Annalisa Convertino, Antonella Lisi, Costantino Del Gaudio

**Affiliations:** 1Institute of Translational Pharmacology, National Research Council, Via Fosso del Cavaliere 100, 00133 Rome, Italy; miriam.merco@ift.cnr.it (M.M.); antonella.lisi@ift.cnr.it (A.L.); 2Institute for Microelectronics and Microsystems, National Research Council, Via Fosso del Cavaliere 100, 00133 Rome, Italy; Antonio.Sciortino@artov.imm.cnr.it (A.S.); annalisa.convertino@cnr.it (A.C.); 3Hypatia Research Consortium, Via del Politecnico snc, 00133 Rome, Italy; elisa.scatena@fondazioneamaldi.it; 4E. Amaldi Foundation, Via del Politecnico snc, 00133 Rome, Italy

**Keywords:** biomimetic scaffolds, bone tissue engineering, regenerative medicine, biocompatibility

## Abstract

The scaffold is a key element in the field of tissue engineering, especially when large defects or substitutions of pathological tissues or organs need to be clinically addressed. The expected outcome is strongly dependent on the cell–scaffold interaction and the integration with the surrounding biological tissue. Indeed, mimicking the natural extracellular matrix (ECM) of the tissue to be healed represents a further optimization that can limit a possible morphological mismatch between the scaffold and the tissue itself. For this aim, and referring to bone tissue engineering, polylactic acid (PLA) scaffolds were 3D printed with a microstructure inspired by the trabecular architecture and biologically evaluated by means of human osteosarcoma SAOS-2 cells. The cells were seeded on two types of scaffolds differing for the designed pore size (i.e., 400 and 600 µm), showing the same growth exponential trend found in the control and no significant alterations in the actin distribution. The microporous structure of the two tested samples enhanced the protein adsorption capability and mRNA expression of markers related to protein synthesis, proliferation, and osteoblast differentiation. Our findings demonstrate that 3D-printed scaffolds support the adhesion, growth, and differentiation of osteoblast-like cells and the microporous architecture, mimicking the natural bone hierarchical structure, and favoring greater bioactivity. These bioinspired scaffolds represent an interesting new tool for bone tissue engineering and regenerative medicine applications.

## 1. Introduction

Bone healing is a naturally occurring process that can be categorized as primary (the fracture gap is less than 0.1 mm and is filled directly by continuous ossification) and secondary (the more common form, which occurs when the fracture edges are less than twice the diameter of the injured bone) [1]. However, the desired outcome is strongly hampered when large injuries occur. Moreover, even if the defect size is not the only parameter used to define a defect as critical, and a commonly agreed treatment is controversial, it has been reported that in most species, a length exceeding 2–2.5 times the diameter of the affected bone can be considered the minimum size that will not heal spontaneously [1,2,3]. Several surgical options are currently available (e.g., autografts, allografts, or synthetic substitutes), but each of them is characterized by advantages and disadvantages that must be critically evaluated by referring to the pathological case to be treated [4]. In this regard, tissue engineering offers an alternative solution by developing ad hoc scaffolds that aim to address all the related issues associated with currently implanted devices, supporting the regeneration of novel autologous tissue. The design of specific scaffolds is, therefore, a key topic that deserves particular attention. The instructive role that it is expected to exert on seeded cells needs to be accurately tailored in order to guide tissue healing and regeneration and to provide an appropriate three-dimensional microenvironment to correctly support biological processes [5]. For this aim, several design requirements should be considered and targeted for the requested therapeutical protocol, including materials, fabrication techniques, addition of bioactive compounds, and surface functionalization. All these cues can be further enhanced by dealing with a scaffold characterized by a microarchitecture resembling that of the biological tissue to be treated. In this respect, and referring to bone pathologies, the preparation of morphological structures similar to the trabecular bone arrangement can be regarded as a functional feature to ensure continuity between the artificial and the natural environment. Such a result can be achieved by means of additive manufacturing thanks to the intrinsic properties to finely control the deposition of the selected materials. Clearly, the resolution can span a wide dimensional range depending on the printing technology, from stereolithography to fused deposition modeling [6], but the possibility to move forward from simple assembled scaffolds, which are unlikely to resemble the natural microarchitecture, can pave the way to a more biomimetic approach to prepare ad hoc tissue-engineered constructs [7]. Several 3D-printed scaffolds have been evaluated for bone tissue engineering as reviewed, for instance, by Wang [8], Zhang [9], and Lin [10], but only a few have been proposed with a trabecular-like architecture [7,11,12,13,14]. The latter approach should be emphasized since the scaffold is supposed to be a temporary extracellular matrix (ECM) and its properties should be replicated, including both bioactive cues and morphological cues. Most scaffolds that have been assessed so far are characterized by a regular structure, which is easily controllable in the design and fabrication stages, allowing prompt regulation of relevant features, such as the porosity, permeability, and mechanical behavior. However, this result can limit the potential of a tissue-engineered construct, especially in vivo, which represents the real “working” environment where any possible drawbacks should be minimized or avoided.

To evaluate the biological response of bone ECM-like scaffolds fabricated by fused deposition modeling (FDM), human osteosarcoma SAOS-2 cells were chosen as the osteoblast model. These cells have an osteoblast phenotype with characteristics that are similar to human primary osteoblasts [15] and, compared to other osteoblast-like cells, better mimic primary human osteoblast cells’ behavior when they interact with biomaterials. Since SAOS-2 cells represent a very good and well-characterized osteoblast-like model, they were used to investigate the in vitro assessment of 3D-printed biomimetic scaffolds to evaluate the biocompatibility, bioactivity, and osteoconductive properties. The effects of two scaffold models, with different microporosity, on cells’ growth, proliferation, metabolic activity, cell morphology, and mRNA expression of key genes, including osteogenic differentiation markers, are reported here.

## 2. Results and Discussion

### 2.1. Adhesion, Growth, and Differentiation of Osteoblast-like Cells on Bioinspired 3D-Printed Scaffolds

The formation, remodeling, and healing of bone is a coordinated process involving various cell types in which osteoblasts, skeletal cells of mesenchymal origin, play a key role as they have the specific ability to synthesize all constituents necessary for bone matrix deposition [16]. Several osteoblast cell lines from different origins have been established as models to study the biology of bone and for in vitro investigation of cell differentiation, cytokine and hormonal regulation, synthesis and secretion of matrix proteins, molecular mechanisms of bone diseases, and drug pharmacokinetics. In the field of tissue engineering, these cell models are extensively used, due to the limited availability of primary human osteoblast cells, for biocompatibility and osteoconductivity evaluation of novel biomaterials. Osteosarcoma cells are among the cell models that are most used in tissue engineering research as they show characteristics of osteoblasts, such as the ability to express specific receptors for 1,25-dihydroxyvitamin D3 and parathyroid hormone, synthetize cell-matrix with alkaline phosphatase (ALP) activity, or produce specific bone matrix proteins [17,18,19]. Although the tumor origin of osteosarcoma cells results in significant phenotypic differences from primary osteoblasts [20] regarding their morphology, proliferation rate, cytokines, growth factors and matrix proteins expression profile, and mineralization activity [21], they are widely used as an in vitro osteoblast-like model for biopharmaceutical evaluations. Human osteosarcoma SAOS-2 cells have an osteoblast phenotype, with higher levels of ALP activity than in other osteosarcoma cell lines [22] that are comparable to human primary osteoblast cells [15]. It has been shown that the SAOS-2 growth factor expression and collagen structure that they produce are similar to those of primary human osteoblast cells [23,24]. More importantly, this cell line, compared to other osteoblast-like cells, can better mimic primary human osteoblast cells’ behavior in response to the biomaterial interaction. For these reasons, and as the human osteosarcoma cell line, SAOS-2, represents a very good and well-characterized osteoblast-like model, it was selected in this study to investigate their in vitro interaction with 3D-printed scaffolds, which mimicked the bone microarchitecture, to assess their biocompatibility and the osteconductive properties.

The cells were seeded on a PLA control case (CTR) and on two different microporous PLA scaffold models (i.e., SCF 400 and SCF 600), and their adhesion, proliferation, and differentiation capability were investigated. First, the cell growth was examined using cycle-phase distribution analysis and the WST-1 colorimetric assay. The propidium-iodide-labeled DNA showed no significant differences in the cell phase percentage of SAOS-2 cells grown for 4 days on both scaffolds compared to the CTR sample, indicating efficient cell cycle activation and progression. These results, as reported in Figure 1, were confirmed by the WST-1 colorimetric assay, which showed a clear exponential trend from day 1 to day 4 in the cells grown on the scaffold surfaces, with the proliferation rate on day 3 and 4 being lower than the CTR cells. This last finding is expected in cells where a differentiation process is on-going, as later confirmed by the expression analysis of osteoblastic markers. The cell growth study was completed by performing nuclei staining analysis with Hoechst 33342, which clearly showed an increase in the number of SAOS-2 cells seeded on the PLA scaffolds from day 1 to day 4 (Figure 1) and an even distribution on the substrate. These results confirmed the lack of cytotoxicity of the PLA scaffolds and indicated that they are a suitable surface for cell attachment.

Using phalloidin staining, the influence of the PLA scaffolds on SAOS-2 cells’ adhesion, flattening, and lengthening was studied, visualizing the actin cytoskeleton morphology and organization. The cytoskeleton, which is essential for maintaining cell shape, is involved in a wide variety of cell functions associated with the differentiation process, including the spatial organization of cell organelles, intracellular membrane traffic, modulation of surface receptors, and the mitosis process [25]. It was observed that the cells attached well to the scaffold substrates and maintained their actin cytoskeleton organization, morphology, shape, size, and orientation, which is similar to the cells grown on CTR (Figure 2).

### 2.2. Differentiation of Osteoblast-like Cells on Bioinspired 3D-Printed Scaffolds

Based on the above-described results, it is possible to assert that the two scaffolds can support SAOS-2 cell adhesion, metabolic activity, and proliferation, providing strong evidence of their biocompatibility and cell-friendly response. To further confirm the 3D-printed scaffolds’ biocompatibility, and to test their osteoconductivity, the mRNA expression of genes involved in osteogenic differentiation or that play a central role in cell biosynthetic machine activity was studied. With this aim, the mRNA expression of the following key constitutive genes was evaluated: β-actin (β-ACT), which is involved in cell motility, structure, and integrity; Ki67, a marker of cellular proliferation; and RPL34, a ribosomal protein involved in cell translation. The mRNA expression of osteogenic markers, such as osteopontin (OPN), alkaline phosphatase (AP), RUNX2, and osteocalcin (OCL), was also evaluated. These mRNA transcripts are highly expressed in SAOS-2 cells and can be considered as health markers of cells. Their expression, which was analyzed in cells grown for 4 days on scaffolds using the RT-PCR assay, was similar among the 2 samples and unchanged when compared to the control (Figure 3). More importantly, the osteogenic commitment capability of the SAOS-2 cells grown on 3D-printed scaffolds was also verified, as a valuable indicator of their capability of maintaining the differentiation potential and consequently their osteoconductive proprieties. PLA is an FDA-approved polymer, which, in addition to having excellent biocompatibility and biodegradability, has osteoinductive properties, including the ability to support the expression of osteogenic markers [26]; therefore, it has been extensively studied for bone repair [27,28].

Osteogenesis is a multistep series of events modulated by an integrated cascade of gene expression characterized by three orderly stages: (1) proliferation; (2) extracellular matrix (ECM) deposition and maturation; and (3) mineralization of the bone ECM [29]. These phases are accompanied by the activation of transcription factors, including RUNX2 and specific genes associated with the osteoblast phenotype, such as alkaline phosphatase (ALP), osteocalcin (OCL), and osteopontin (OPN). ALP is considered a marker of early differentiation, which is expressed at the end of the proliferative period and during deposition and maturation of the ECM. Both OPN, the main phosphorylated glycoprotein of bone, and OCL, a highly conserved small molecule, are expressed during the last phase of osteogenesis, which is associated with mineralization of the bone matrix [30]. mRNA expression analysis of these osteogenic markers was conducted in cells grown for 4 days on the PLA scaffolds (Figure 4) using the qRT-PCR assay, revealing a statistically significant increase in the 4 markers for both samples (i.e., SCF 400 and SCF 600) compared to those grown on the control. The collected results highlight that the osteogenic phenotype is not only maintained in cells grown on the PLA scaffolds, confirming the well-established osteoinductive properties of PLA [31], but that the differentiation process was efficiently activated and upregulated in the SCF 400 and SCF 600 microporous 3D scaffolds, as shown by the higher levels of early and late bone differentiation markers expressed in both 3D PLA scaffolds compared to CTR.

### 2.3. Bioactivity of 3D-Printed Scaffolds

It is hypothesized that the positive effect of the scaffolds on SAOS-2 differentiation is related to their bioinspired 3D microarchitectures and microporosity, which mimics the natural extracellular matrix of the bone and favors greater bioactivity. Together, these two features contribute to the creation of an osteoinductive microenvironment and stimulate osteogenic differentiation. In support of this conclusion, many studies have reported that microporosity plays a significant role in enhancing the osteoinduction of 3D scaffolds [32]. The microporosity increases the surface area of the scaffold, providing more protein adsorption sites, which facilitate the interactions between supports and cells and improve the nutrient’s availability [33]. The absorbed proteins can subsequently stimulate the osteogenic-related functions of cells, such as attachment, proliferation, osteogenic differentiation, and biomineralization [34,35]. Moreover, the capillary force generated by the microporosity improves the attachment of cells on the scaffolds’ surface and enables cell penetration into micropores that are smaller than them. To deeply study the adhesion and penetration capability of cells in the scaffolds, their microstructure was investigated, after seeding SAOS-2 cells, using scanning electron microscopy (SEM). The images of both samples reported in Figure 5 confirm a random porous microarchitecture, in which the microporosity is very evident, which is different from the flat and ordered structure of the CTR. Furthermore, the cells appear to be well attached to the scaffold surfaces, confirming the confocal analysis, and many of them are concentrated around the pore wall, demonstrating an ability to penetrate the scaffold.

As discussed, the capability of biomaterials to absorb protein is directly related to their bioactivity and impact on the interaction between the cells and scaffolds, playing a key role in proliferation and osteogenic induction. Proteins are adsorbed before cells adhere to the surface of biomaterials and the kinetics of this process can influence the subsequent cell behavior. In this study, to examine the scaffold bioactivity related to the protein adsorption capability, the kinetics of protein adsorbed on the scaffold surfaces was measured at various incubation intervals (1, 2, 3, 4, and 7 days) and compared to CTR. As shown in Figure 6, the amount of proteins adsorbed by the two tested scaffolds was higher compared to CTR at all time points. Both scaffolds show a rapid increase on day 1 followed by a further noticeable increase in protein adsorption on SCF 600 and a slight but significant increase on SCF 400 compared to CTR. The collected data suggest that the microarchitecture of both scaffolds, characterized by a greater porosity, enhanced the protein adsorption capacity on SCF 600, which has a major porosity compared to SCF 400, and thus improved the bioactivity.

## 3. Materials and Methods

### 3.1. Scaffold Fabrication

The design and fabrication procedures of the 3D-printed scaffolds have already been reported [5]. Briefly, 2 random 3-dimensional distributions of spheres, i.e., virtual pores, with a diameter of 400 and 600 μm, were created using a custom-made script and subtracted from a box-shaped volume (10 × 10 × 3 mm) to produce porous CAD models. These models were then imported in ideaMaker (Raise3D Inc., Irvine, CA, USA) and sliced at 0.25 mm in the Z direction. Scaffolds were fabricated by processing polylactic acid (PLA filament; Formfutura BV, The Netherlands) using a Raise 3D N2 printer (Raise 3D Inc., Irvine, CA, USA), setting the nozzle temperature at 205 °C and that of the build surface at 40 °C. According to the design procedure, the 3D-printed scaffolds were labeled SCF 400 and SCF 600, respectively. As a control case, box-shaped PLA samples with a flat surface were fabricated.

### 3.2. Cell Culture

The human osteosarcoma SAOS-2 cell line was obtained from the American Type Culture Collection (ATCC, HTB-37 Rockvile, MD, USA). The cells were grown in high-glucose Dulbecco’s modified Eagle’s Medium (DMEM; Euroclone, Milan, Italy), supplemented with 10% heat-inactivated fetal bovine serum (FBS, Euroclone), 2 mM L-glutamine (Sigma, Darmstadt, Germany), 1.0 unit/mL penicillin (Sigma), and 1.0 mg/mL streptomycin (Sigma, Darmstadt, Germany). The cells were cultured on a plastic Petri dish at 37 °C in a humidified incubator containing 5% CO_2_. For all experiments, the scaffolds were immersed in ethanol 70% for 30 min, and washed with phosphate-buffered saline (PBS). SAOS-2 cells (3 × 10^4^ cells/cm^2^) were seeded on both the PLA scaffolds’ surfaces and on the PLA sample (CTR) and grown for up to 4 days.

### 3.3. Cell Growth Analysis

The SAOS-2 cell growth trend was quantified by a colorimetric assay based on oxidation of water-soluble tetrazolium salts (Cell Proliferation Reagent WST-1; Roche Diagnostics, Darmstadt, Germany). Exponentially growing SAOS-2 cells were seeded on both scaffold surfaces and on the PLA sample (CTR) at a density of 3 × 10^4^ cells/cm^2^, cultured for up to 4 days in a humidified incubator (37 °C, 5% CO_2_), and analyzed every day. WST-1 reagent diluted to 1:10 was added to the medium of SAOS-2 cells on day 1, 2, 3, and 4 following plating, and after an incubation of 2 h in a humidified atmosphere, the supernatants (100 µL) of cells were placed in 96-well plates and analyzed using the formazan dye. Quantification of the produced formazan dye was performed by absorbance measurement at 450 nm with a scanning multiwell spectrophotometer (Biotrack II; Amersham Biosciences, Little Chalfont, UK). For the nuclei analysis, cells were washed three times with PBS, stained for nuclei localization with Hoechst 33,342 (trihydrochloride–trihydrate), and examined.

### 3.4. Cell Cycle Analysis by Flow Cytometry

The cell cycle of SAOS-2 cells grown for 4 days was analyzed by flow cytometry analysis. A single-cell suspension of cells was fixed in 10 mL of 70% cold ethanol at 4 °C. Fixed cells were washed in PBS, then stained with propidium iodide (20 µg/mL; Sigma) and RNase A (250 µg/mL; Sigma) solution for 30 min at room temperature in the dark [36]. Approximately 1 × 10^6^ cells were acquired using an FACSCalibur (Becton Dickinson) cytometer and the cell cycle analysis was performed using ModFIT LT 2.0 software.

### 3.5. Real-Time Quantitative RT-PCR Analysis

Total RNA was extracted from the SAOS-2 cells, grown on both the PLA sample and the scaffold surfaces for 4 days, using TRIzol Reagent (Invitrogen). One microgram of total RNA was used to synthesize first-strand cDNA with random primers, using an iScriptTM cDNA synthesis kit (Bio-Rad, CA, USA). Quantification of all gene transcripts was carried out by real-time quantitative polymerase chain reaction (RT-qPCR), using the SsoAdvanced™ Universal SYBR^®^ Green Supermix (Bio-Rad) and Bio-Rad Real-Time PCR Detection Systems. Each reaction was run in triplicate and contained 0.5 µL of cDNA template along with 250 nM primers in a final reaction volume of 20 µL. The investigated genes are reported in Table 1. The cycling parameters were 50 °C for 2 min, then 95 °C for 10 min to activate DNA polymerase, then 40 cycles of 95 °C for 15 s, and finally, 60 °C for 1 min. The melting curves were produced using Dissociation Curves software (Bio-Rad) to ensure that only a single product was amplified. As negative controls, tubes where RNA or reverse transcriptase were omitted during the RT reaction were used. Experiments were carried out to compare the relative levels of each transcript and endogenous control GAPDH in every sample. The data were analyzed using the following equation described by Livak [36]:Amount of target was calculated using the 2^−ΔΔCt^ equation.
ΔCt = (average target Ct − average GAPDH Ct)
ΔΔCt = (average ΔCt treated sample − average ΔCt untreated sample)

Before using the ΔΔCt method for quantification, a validation experiment was performed to demonstrate that the efficiency of the target genes and the reference GAPDH was equal.

### 3.6. Scanning Electron Microscopy Analysis

The cells plated on the scaffolds were fixed with 4% paraformaldehyde for 10 min and the dried specimens were coated with an evaporated Au thin film (10 nm) before analysis with a ZEISS SIGMA 300 field emission SEM. The morphological analysis was verified at an accelerating voltage of 5 kV using a secondary electron (SE) detector.

### 3.7. Confocal Laser Scanning Microscopy Analysis

SAOS-2 cells seeded on scaffolds and CTR, and grown for 48 h, were fixed with 4% paraformaldehyde for 10 min and permeabilized with 0.2% triton X-100 in PBS containing 1% bovine serum albumin for 5 min. The cells were then incubated with phalloidin tetramethylrhodamine isothiocyanate conjugated (1:100) with an anti-actin toxin (Sigma) in a blocking buffer for 1 h, washed 3 times in PBS/BSA, and stained for nuclei localization with Hoechst 33342. Cover slips were collected, cell-side down, on a microscope slide with 0.625% N-propyl gallate in PBS glycerol 1:1. The cover slip ‘sandwich’ was sealed to prevent exposure to air and to exclude and prevent the crystal formation of H_2_O. Fluorescence analyses were performed using a LEICA TCS 4D Confocal Microscope supplemented with an Argon Krypton laser and equipped with 40 × 1.00–0.5 and 100 × 1.3–0.6 oil immersion lenses. Confocal optical Z sections were acquired at 2-μm intervals for each field considered and a middle confocal Z section is shown.

### 3.8. Protein Adsorption

Protein adsorption was determined using the method of Bradford. Each scaffold was incubated in a 24-well plate with 1 mL of fetal bovine serum (FBS) at 37 °C. The concentration of the protein in the FBS solution was measured with a commercial protein assay kit (Biorad). Samples were immersed in a fetal bovine serum (FBS) solution and at various incubation intervals (1, 2, 3, 4, and 7 days). The amount of proteins adsorbed was calculated by subtracting the amount of proteins left in the FBS solution after adsorption from the amount of proteins in the control FBS solution (without sample) under the same incubation conditions.

### 3.9. Statistics Analysis

Each experiment was performed three times (*n* = 3). The results are presented as mean ± SD. The significance of the difference was evaluated using the Student’s *t* test, with *p* < 0.05 as the minimum level of significance.

## 4. Conclusions

The use of 3D-printed scaffolds that resemble the microarchitecture of bone tissue can pave the way for an improved tissue engineering approach to treat several pathologies and enhance the expected outcome. In this respect, the biological response of the present tested scaffolds proved a clear biocompatibility and cell-friendly properties and, more importantly, the capability to induce/promote and support osteogenic differentiation, which is directly correlated with the bone-like architecture and porous microstructure, making them biomimetic and bioactive.

## Figures and Tables

**Figure 1 ijms-23-05383-f001:**
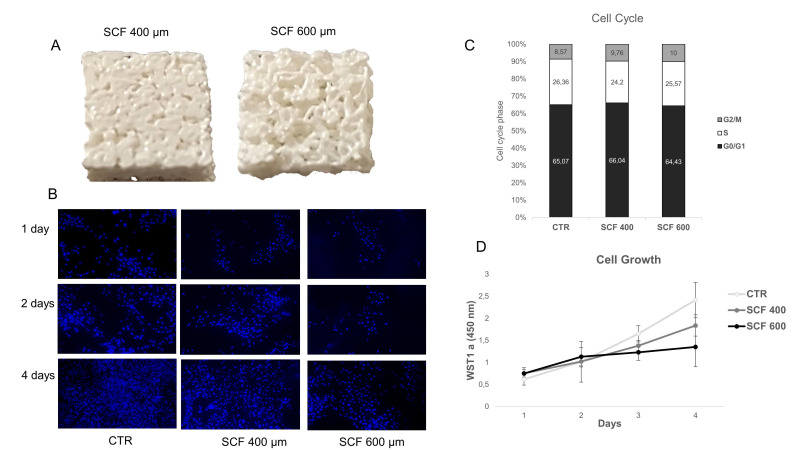
(**A**) 3D-printed scaffolds with a pore size of 400 (**left**) and 600 µm (**right**). (**B**) Time evolution of SAOS-2 nuclei on SCF 400 scaffolds, SCF 600 scaffolds, and PLA control (CTR), revealed by Hoechst staining 33342, magnification 10× (**C**) Cell cycle analysis of SAOS-2 seeded on SCF 400 scaffolds, SCF 600 scaffolds, and PLA control (CTR). (**D**) Cell growth analysis using the WST-1 assay of SAOS-2 seeded on SCF 400, SCF 600, and PLA control (CTR) (data are shown as mean SD).

**Figure 2 ijms-23-05383-f002:**
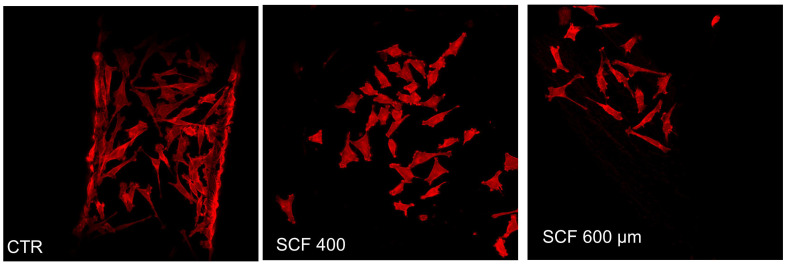
Actin distribution analysis of the SAOS-2 cells seeded on SCF 400 scaffolds, SCF 600 scaffolds, and PLA control (CTR) by confocal laser scanning microscopy.

**Figure 3 ijms-23-05383-f003:**
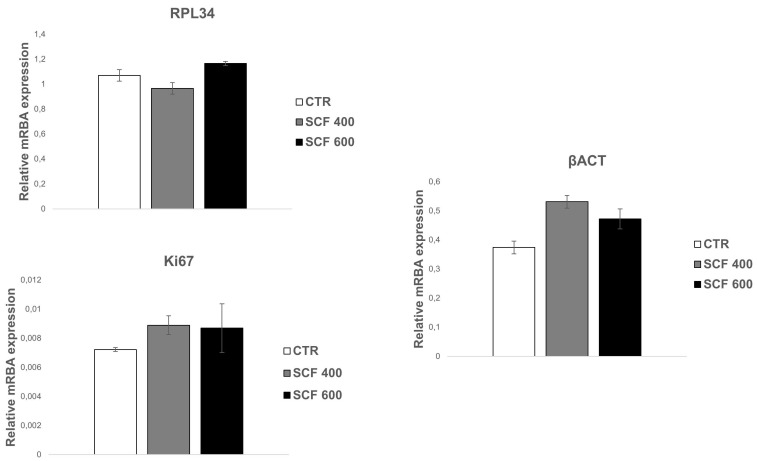
qRT-PCR analysis of SAOS-2 key gene expression. The housekeeping genes (constitutive gene) β-ACT, Ki67, and RPL34 were investigated in SAOS-2 cells grown on SCF 400 scaffolds, SCF 600 scaffolds, and compared to PLA control (CTR). Data are shown as mean SD.

**Figure 4 ijms-23-05383-f004:**
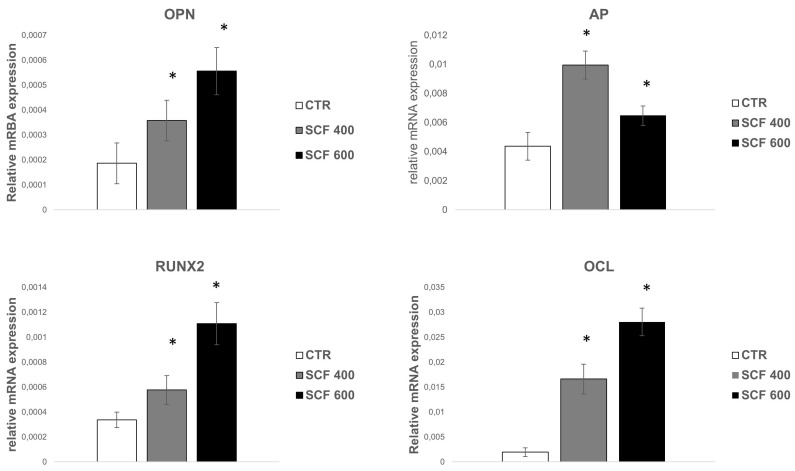
qRT-PCR analysis of the early and late osteoblast differentiation markers’ (RUNX2, AP OPN, and OCL) expression on SAOS-2 cells grown on SCF 400 and SCF 600 scaffolds compared to PLA control (CTR). * identify statistical significance (*p* < 0.05). Data are shown as mean SD.

**Figure 5 ijms-23-05383-f005:**
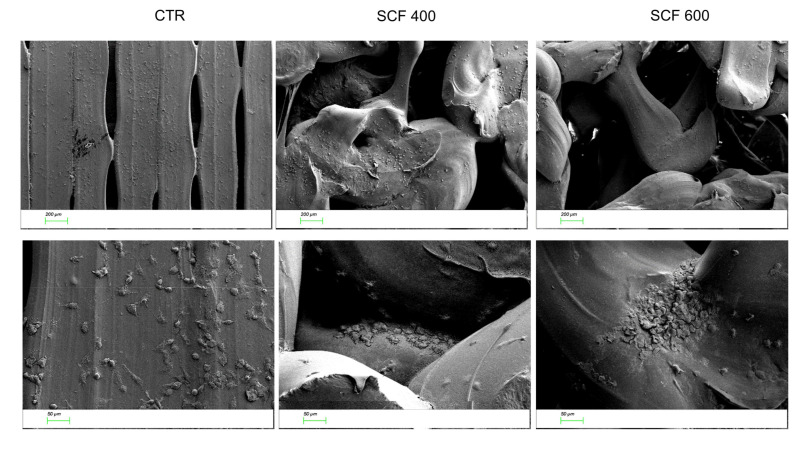
SEM images of the cells seeded on the CTR (**left panel**), SFC 400 (**central panel**), and SFC 600 (**right panel**) scaffolds at 2 different magnifications (50× top line, 200× bottom line).

**Figure 6 ijms-23-05383-f006:**
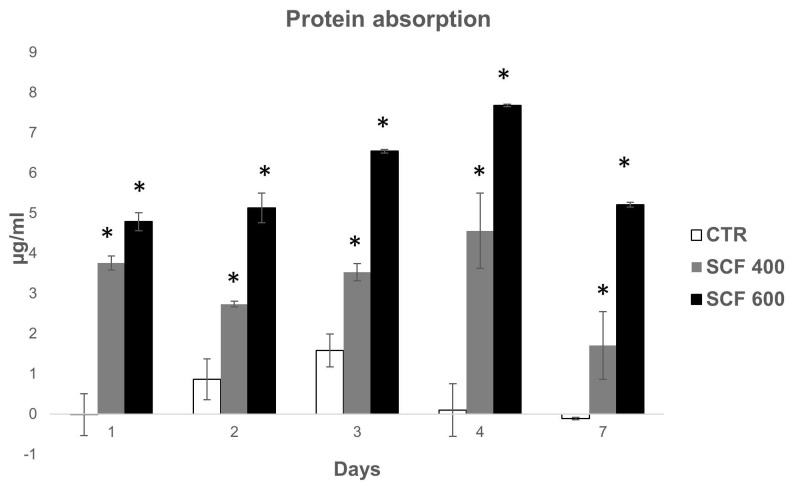
Protein absorption of the SCF 400 and SCF 600 scaffolds compared to the PLA control. Data are shown as mean SD. * identify statistical significance (*p* < 0.05).

**Table 1 ijms-23-05383-t001:** Sequence of primers used for qRT-PCR.

Target Gene	Primer Sequence	Annealing Temperature (°C)
β-ACT	5′-gctcctcctgagcgcaag-3′ 5′catctgctggaaggtggaca-3′	60
OPN	5′-gtgtggtttatggactgagg-3′ 5′-acggggatggccttgtatg-3′	60
Ki67	5′-tgaacaaaaggcaaagaagac-3′ 5′-gagctttccctattattatggt-3′	60
RPL34	5′-gaaacatgtcagcagggcc-3′ 5′-tgactctgtgcttgtgcctt-3′	60
RUNX2	5′-catcatctctgccccctct-3′ 5′-actcttgcctcgtccactc-3′	60
ALP	5′-caatgagggcaccgtggg-3′ 5′-tcgtggtggtcacaatgcc-3′	60
OCL	5′-cagcgaggtagtgaagag-3′ 5′-gaaagccgatgtggtcagc-3′	60
GAPDH	5′-catcatctctgccccctct-3′ 5′-caaagttgtcatggatgacct-3′	60

## Data Availability

Not applicable.

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
