# Peer review of "Biological Response to Bioinspired Microporous 3D-Printed Scaffolds for Bone Tissue Engineering"

_ijms, 2022, doi:10.3390/ijms23105383_

Round 1
Reviewer 1 Report
This paper tackles a quite classical and largerly discussed and published topic in tissue engineering, i.e. the use of biomimetic scaffolds for osteoinduction and bone repair. The experience of the authors is focused on polylevolactic acid (PLLA), a thermoplastic biomaterial with well studied osteoinductive properties and mechanical features tested in a number of different bone experimental settings including animal models and man. The results are quite in line with what currently known on PLLA role in in vitro osteodifferentiation and growth of osteogenic cells, but a number of points should preliminarly be introduced and discussed more cogently by the authors:
ABSTRACT
The authors correctly underline the key role of the natural ECM for bone healing and prevention of morphological mismatch. However, what exactly do they mean for “mimicking”? and what should be the mismatch? In other words, which characteristics of the ECM do they intended useful / needed to be mimicked? And, again, what would be the two biological / structural terms of the supposed mismatch? This is relevant to the paper’s title where emphasis is given to the term “morphology” of the scaffold, though a number of microarchitectural features are involved in the activity of the ECM. Please, clarify.
The authors state that a random microstructure like that ensuing from their fabrication procedure would replicate the bone trabecular architecture. This concept is repeated in two different sections of the Abstract, stressing that a random microarchitecture mimicks the natural bone ECM. However, there is a well known anatomical, ultrastructural and even radiological awareness in the international literature that bone trabecular architecture and related ECM arrangement are not at all random in any human bone. In fact, bone morphofunctional units, and related extracellular ECM natural scaffolds have a very specific geometrical arrangement depending on the bone part (compact vs cancellous bone), type of bone (long vs short vs flat bones), and sector of the body where the bone is located (head vs trunk vs arms). Indeed, bone microarchitecture comprising all its components like ECM dictates the mechano-chemical constraints influencing the activity of resident cells and their relation to the mineral mass, making possible to identify cellular and mineral microdomains in the bones wih selected activities. Please, show knowledge of this and justify the choice of a random architecture of the scaffold in the experiments presented.
The authors talk of an exponential trend, and make comparison with the control. What kind of exponential trend do the authors refer to? Cell growth? Cell differentiation? Other? Please, clarify.
INTRODUCTION
The authors open their presentation by sentencing that bone healing is a natural process for repairing small defects. Although their enter and quote later some specifications on this restriction, however it should be clearly stated that bone healing is a general process of repair that occurs in any kind of bone defect including large bone lesions, though its efficiency depends on the defect size having large defects a more disordered process than small ones. Please, better focus the sentence.
RESULTS AND DISCUSSION
An incorrect use of the English verbs and punctuation emerges through lines 120 - 124. Please amend.
As a matter of fact, then, the authors emphasize the enhanced response of osteogenic markers in the grown cells, but it seems they scarcely valorize the international literature largely dealing with the well established osteinductive properties of PLLA per se including its capacity to enhance expression of osteogenic markers.
In addition, they state that the positive effects of the scaffold on osteogenic differention would be related to the bioinspired 3D random microarchitecture mimicking the natural ECM. Again, as raised and detailed in the Abstract section this concept should be substantially reanalyzed and blunt in light of the evidence highlighted above. However, the reviewer fully agrees with the concept and role of microporosity, that seems to him the major achievemet of the work and should be brough to the title as opposed to that more generical of “morphology” of the scaffold implying a much more complex microarchitectural three-dimensional setting with respect to that analyzed by the authors.
Author Response
Response to Reviewer 1 Comments
This paper tackles a quite classical and largerly discussed and published topic in tissue engineering, i.e. the use of biomimetic scaffolds for osteoinduction and bone repair. The experience of the authors is focused on polylevolactic acid (PLLA), a thermoplastic biomaterial with well studied osteoinductive properties and mechanical features tested in a number of different bone experimental settings including animal models and man. The results are quite in line with what currently known on PLLA role in in vitro osteodifferentiation and growth of osteogenic cells, but a number of points should preliminarly be introduced and discussed more cogently by the authors:
ABSTRACT
The authors correctly underline the key role of the natural ECM for bone healing and prevention of morphological mismatch. However, what exactly do they mean for “mimicking”? and what should be the mismatch? In other words, which characteristics of the ECM do they intended useful / needed to be mimicked? And, again, what would be the two biological / structural terms of the supposed mismatch? This is relevant to the paper’s title where emphasis is given to the term “morphology” of the scaffold, though a number of microarchitectural features are involved in the activity of the ECM. Please, clarify.
Response: The Authors thank the Reviewer for focussing on the “mimicking” concept which needs to be further clarified for better defining the field of investigation of the proposed study. In order to provide a different cell platform from those generally fabricated by means of 3D technologies, in which a regular pattern is assembled and investigated, a “random” structure is here developed. In this regard, the design approach is aimed to offer a bone tissue-like microenvironment to the seeded cells comparable to the physiological one. This should ensure a more uniform transition from the scaffold architecture to the bone structure, thus avoiding a possible morphological mismatch that can affect the cell response (it should be noted that a possible mismatch can be also related to the mechanical properties, being in turn affected by the material and the three-dimensional arrangement of the structural elements as well). Therefore, answering to the question “which characteristics of the ECM do they intended useful / needed to be mimicked?”, the easiest reply is “A tissue engineered scaffold should mimic all the ECM characteristics”. However, such an answer would be unrealistic and disrespectful, and the point of view of the Authors is to optimize the most instructive features to deal with a functional tissue engineered scaffold. For this aim, the microarchitecture can be regarded as the first one to be tailored before to move forward and provide, for instance, a biochemical modification to add more ECM-like cues.
Regarding the question “what would be the two biological/structural terms of the supposed mismatch?”, it is not clear what are the “two biological/structural terms” the Reviewer is referring to. Maybe, this request of clarification is focussed on the two different pore sizes considered in the study, i.e., 400 and 600 µm. If so, in order to optimize the design of the scaffolds, the Authors tested two different conditions to infer on the role of pore size, being pivotal for promoting the expected scaffold performance thanks to the beneficial impact to improve, for instance, mass transport and cell migration within the scaffold, as a consequence.
The authors state that a random microstructure like that ensuing from their fabrication procedure would replicate the bone trabecular architecture. This concept is repeated in two different sections of the Abstract, stressing that a random microarchitecture mimicks the natural bone ECM. However, there is a well known anatomical, ultrastructural and even radiological awareness in the international literature that bone trabecular architecture and related ECM arrangement are not at all random in any human bone. In fact, bone morphofunctional units, and related extracellular ECM natural scaffolds have a very specific geometrical arrangement depending on the bone part (compact vs cancellous bone), type of bone (long vs short vs flat bones), and sector of the body where the bone is located (head vs trunk vs arms). Indeed, bone microarchitecture comprising all its components like ECM dictates the mechano-chemical constraints influencing the activity of resident cells and their relation to the mineral mass, making possible to identify cellular and mineral microdomains in the bones wih selected activities. Please, show knowledge of this and justify the choice of a random architecture of the scaffold in the experiments presented.
Response: From the concern raised by the Reviewer, the Authors think that a misunderstanding occurred. The Authors are more than aware that bone architecture is strictly dependent on the bone part, type and location. It is clear that all the properties of each bone segment, including morphological, mechanical and functional ones, are strictly correlated and the modification of one of them induce an adaptive response on the others: no doubt about that. As previously reported, the Authors used the term “random” just to differentiate their scaffolds from those assembled with a regular and repetitive pattern that cannot be representative of the bone physiological architecture. To be definitely biomimetic, a scaffold should accurately reproduce the micro-morphology of the tissue to be healed/replaced, as a starting point, and to accomplish this goal the most suitable option is to deal with clinical data from, e.g., CT scans to be used as input files to fabricate an ad hoc and personalized scaffold. However, as the Reviewer may agree, these data are not always available and alternative off-the-shelf solutions should be therefore developed. Following this latter approach, the proposed study presents a possible methodology to prepare bone tissue-like scaffolds to be tested for tissue engineering applications.
Agreeing with the Reviewer, the incidence of the “random” concept was already clearly defined and circumscribed in the submitted version as in the Introduction section it was reported “All these cues can be further enhanced dealing with a scaffold characterized by a microarchitecture resembling that of the biological tissue to be treated. In this respect, and referring to bone pathologies, the preparation of morphological structures similar to the trabecular bone arrangement can be regarded as a functional feature to ensure continuity between the artificial and the natural environment” and “…but the possibility to move forward from simple assembled scaffolds, unlikely resembling the natural microarchitecture, can pave the way to a more biomimetic approach to prepare ad hoc tissue engineered constructs [7]”.
However, thanks to the Reviewer’s considerations, the “random” concept has been revised/mitigated throughout the manuscript.
Finally, considering the previous reply and this one, and the following suggestion in the Results and Discussion section, the title of the paper has been modified as follows “Biological response to bioinspired microporous 3D printed scaffolds for bone tissue engineering”.
The authors talk of an exponential trend, and make comparison with the control. What kind of exponential trend do the authors refer to? Cell growth? Cell differentiation? Other? Please, clarify.
Response: Many thanks for your observation, we are referring to a cell growth trend. It was added, accordingly.
INTRODUCTION
The authors open their presentation by sentencing that bone healing is a natural process for repairing small defects. Although their enter and quote later some specifications on this restriction, however it should be clearly stated that bone healing is a general process of repair that occurs in any kind of bone defect including large bone lesions, though its efficiency depends on the defect size having large defects a more disordered process than small ones. Please, better focus the sentence.
Response: The Authors agree with the Reviewer and the Introduction has been modified as follows
“Bone healing is a natural occurring process that can be categorized as primary (the fracture gap is less than 0.1 mm and is filled directly by continuous ossification) and secondary (the more common form and occurs when the fracture edges are less than twice the diameter of the injured bone) [1]. However, the desired outcome can be strongly hampered when dealing with large injuries and, even if the defect size is not the only parameter to define a defect as critical and a commonly agreed treatment can be controversial, it has been reported that in most species a length exceeding 2-2.5 times the diameter of the affected bone can be considered the minimum size that will not heal spontaneously [1; De La Vega et al., 2022; Roohani et al., 2021].”
The following references were added:
De La Vega RE, van Griensven M, Zhang W, Coenen MJ, Nagelli CV, Panos JA, Peniche Silva CJ, Geiger J, Plank C, Evans CH, Balmayor ER. Efficient healing of large osseous segmental defects using optimized chemically modified messenger RNA encoding BMP-2. Sci Adv. 2022;8(7):eabl6242. doi: 10.1126/sciadv.abl6242.
Roohani I, Yeo GC, Mithieux SM, Weiss AS. Emerging concepts in bone repair and the premise of soft materials. Curr Opin Biotechnol. 2021;74:220-229. doi: 10.1016/j.copbio.2021.12.004. Epub ahead of print.
RESULTS AND DISCUSSION
An incorrect use of the English verbs and punctuation emerges through lines 120 - 124. Please amend.
Response: It was done, accordingly.
As a matter of fact, then, the authors emphasize the enhanced response of osteogenic markers in the grown cells, but it seems they scarcely valorize the international literature largely dealing with the well established osteinductive properties of PLLA per se including its capacity to enhance expression of osteogenic markers.
Response: Thank you for the appropriate and useful suggestion. We have better described the osteoinductive properties of PLA, including its ability to enhance expression of osteogenic markers, and highlighted the SCF 400 and SCF 600 3D printed scaffolds effect.
In addition, they state that the positive effects of the scaffold on osteogenic differention would be related to the bioinspired 3D random microarchitecture mimicking the natural ECM. Again, as raised and detailed in the Abstract section this concept should be substantially reanalyzed and blunt in light of the evidence highlighted above. However, the reviewer fully agrees with the concept and role of microporosity, that seems to him the major achievemet of the work and should be brough to the title as opposed to that more generical of “morphology” of the scaffold implying a much more complex microarchitectural three-dimensional setting with respect to that analyzed by the authors.
Response: In accord with the reviewer's suggestion, as reported above, we have better explained the concept of bone tissue microarchitecture. Additionally, the title of the paper has been changed considering the role played by microporous architecture on the biological response to the 3D printed scaffold reported by many studies that demonstrate a significant osteoinductive effect of microporosity, as described in subsection 2.3 of "result and discussion". The microporosity improves the bioactivity of the scaffolds, with increased protein absorption that facilitates interactions between supports and cells. This helps to stimulate the osteogenic functions of the cells, including osteogenic differentiation. Following the reviewer's suggestion, the concept of microporosity was highlighted even more at various points in the article.
Reviewer 2 Report
//
The manuscript entitled “Biological response to morphologically biomimetic 3D printed scaffolds for bone tissue engineering” is presenting the data of a thorough in vitro study on the effect of the microarchitecture of a scaffold for hard tissue engineering on its biological behavior. The results are well argued (colorimetric assay and flow cytometry analysis for cell growth investigation, Real-Time quantitative RT-PCR analysis for quantification of 8 gene transcripts, completed by Confocal Laser Scanning Microscopy and Scanning Electron microscopy observation, protein adsorption evaluation). There are other literature data on this theme (random and porous microstructure effect on biological response of a printed scaffold- cell adhesion, proliferation as well as the expression of specific osteogenic genes), , as specified in Reference chapter, but here the study is more comprehensive, clear, well organized. It is useful especially for the development of biomimetic, efficient scaffolds, with appropriate design using additive manufacturing technology/bioprinting, pointing on the importance of implementing different criteria to guide the microstructural design of a scaffold.
Observations:
- The phrase from page 6, lines 218-222 must be moved to experimental part to complete 3.8 Protein Adsorption
- Maybe the introduction in the text of a representation of scaffolds design or a microphotograph will be useful.
Round 2
Reviewer 1 Report
Authors correctly addressed a substantial part of the criticism and suggestions, and the reviewer believes that now the manuscript is definitely ameliorated. The reviewer congratulate for the wise choice of reshaping the title, now much more consistent with the scaffold model, and results obtained.